# Critical Care Management of Surgically Treated Gynecological Cancer Patients: Current Concepts and Future Directions

**DOI:** 10.3390/cancers17152514

**Published:** 2025-07-30

**Authors:** Vasilios Pergialiotis, Philippe Morice, Vasilios Lygizos, Dimitrios Haidopoulos, Nikolaos Thomakos

**Affiliations:** 1First Department of Obstetrics and Gynecology, Division of Gynecologic Oncology, “Alexandra” General Hospital, National and Kapodistrian University of Athens, 10679 Athens, Greece; vlygizos@gmail.com (V.L.);; 2Gustave-Roussy Cancer Campus, Department of Gynecologic Surgery, 94800 Villejuif, France; 3University Paris-Saclay, 94270 Le Kremlin-Bicêtre, France

**Keywords:** critical care, high dependency unit, intensive care unit, gynecological cancer

## Abstract

Critical care management in surgically treated gynecological cancer patients has not been investigated yet, despite the fact that it may considerably help improve patient outcomes. Given the significant impact that the advancements made in the last 20 years in the treatment of gynecological cancer patients have on patient survival, it seems reasonable to discuss the need for innovative strategies that will target an intensive management of patients with various medical conditions that are precluded from optimal treatment plans due to comorbidities that were previously considered contraindications for aggressive management. The present review discusses how critical care management should be structured in gynecologic oncology and how it can help ameliorate outcomes of surgically treated patients.

## 1. Introduction

Gynecological cancer incidence and mortality increases overtime, constituting approximately 40% of cancer cases in women and accounting for 30% of cancer related deaths according to the findings of the latest study that is based on data from the Surveillance, Epidemiology, and End Results (SEER) program [1]. This observation may be the result of the advancing age of the general population, which directly affects the survival outcomes of cancer patients [2]. Considering this, it seems reasonable to assume that, leaving aside tumor biology and the late referral of elderly patients, the presence of various comorbidities affects the actual perioperative outcome as well as the medical treatment that these patients receive [3]. Therefore, elderly patients with cervical cancer are more likely to receive radiotherapy than undergo radical surgery in the early stages of the disease, whereas patients with ovarian cancer will more likely undergo fewer radical procedures to limit the possibility of severe postoperative complications.

Considering the advances that have been achieved the last 20 years in the treatment of gynecological cancer patients and their considerable effect on patient survival, it seems reasonable to communicate the need for novel strategies that will target an intensive management of patients with various medical conditions that are precluded from optimal treatment plans. The value of critical care units seems to be particularly promising as they provide the means for appropriate perioperative patient management.

In the present review, we summarize the major organ alterations that are expected to occur in gynecological cancer surgery and provide evidence concerning the actual contribution of intensive care facilities in the management of these women.

## 2. Review Methods

The present narrative review is based on evidence focusing on critical care management in cancer patients. Studies focusing on gynecological cancer patients were expected to be extremely limited, therefore, the extrapolation of outcomes and recommendations from generalized populations suffering from cancer were retrieved when these were not explicitly addressed in gynecologic oncology groups. Scientific papers published in the English language were selected, with priority being given to clinical trials, when available, and prospective cohort studies. In the absence of relevant data, retrospective cohorts and case–control studies were considered. The search was focused but not limited to the latest evidence, namely, research that expanded from 2015 and thereafter.

## 3. Major Physiological Changes Supporting Critical Care Support

Cancer itself is a condition that is extremely dynamic for the human body. The rapid tumor growth as well as apoptosis due to systemic treatment increase the demands of several organs and systems within the human body. When the capabilities of the human physiology are surpassed, organ dysfunction is encountered and rigorous patient management is required to prevent organ failure, which may have a detrimental effect on the actual course of the disease, therefore directly affecting patient survival. Common factors affecting the decision to admit a gynecological cancer patient in critical care are summarized in Figure 1 and described hereinafter.

### 3.1. Cardiovascular System

Cardiovascular disease is rather common in cancer patients with an incidence that ranges between 12 and 60% of cases [4]. The intersection of obesity, gynecological cancer and cardiovascular disease has been previously described and the correlation seems to be amphidromous as cardiovascular disease has been associated with an increased risk of developing cancer and vice versa [5,6,7,8]. The impact of pre-existing cardiovascular morbidity on gynecological cancer survival has also been described with several studies indicating that ovarian cancer patients as well as endometrial cancer patients are directly affected by the presence of this comorbidity [9,10,11,12].

It is important to differentiate patients with a pre-existing heart condition from those that develop significant cardiovascular alterations that affect the perioperative course and result in significant morbidity. Patients with pre-existing cardiovascular morbidity are most likely obese and at the advanced stage and they are easily identified during the preoperative assessment, whereas a significant proportion of patients that develop cardiovascular pathology due to toxicity from cancer-related treatments are younger in age and subject to multiple therapeutic modalities that may result in a series of adverse effects, including hypertension, myocardial ischemia, arrythmias, cardiomyopathy and even coronary artery disease and heart failure that may be wrongly misidentified [13,14].

The impact of major abdominal surgery on the cardiovascular system is detrimental, and a recent multi-institutional study that was based on outcomes that were retrieved from a prospective database of 24,000 patients indicated that approximately 1 in 4 patients that die in the immediate postoperative period (30 days) develop significant cardiovascular complications [15]. Perioperative myocardial injury is an established factor of mortality in non-cardiac surgery, even among cases that are diagnosed early [16]. However, it should be noted that the majority of cases that develop myocardial injury after non-cardiac surgery do not have clinical signs or symptoms and are only manifested by transient fluctuations in troponin values [17]. Even these subclinical manifestations may, however, affect the survival outcomes of these patients, as current evidence suggests that cardiac troponins are independently associated with the survival outcomes of cancer patients [18]. The underlying pathophysiology of perioperative myocardial injury seems to differ from that of patients that develop spontaneous myocardial injury. Two potential distinct groups of patients have been reported, the first being patients that develop acute coronary syndrome following rupture, fissuring or erosion of a vulnerable plaque [19]. The process is thought to be the result of stress and increased excretion of catecholamines and cortisol following major abdominal surgery that may result in tachycardia and hypertension, which exerts shear stress in unstable plaques [20]. The second type of perioperative myocardial injury refers to patients that develop a significant myocardial oxygen supply–demand imbalance that is manifested as silent ST-segment abnormalities [19]. These may span a spectrum of minor, transient and silent ST-segment depressions that are linked to low-level troponin elevations, up to overt ischemia that is linked to marked troponin elevation [21]. The underlying causes of myocardial oxygen supply–demand imbalance are numerous; however, the predominant reasons involve severe perioperative hypotension due to hypovolemia, bleeding or vasodilation, hypertension that develops from vasoconstriction as well as abnormal blood gasses (primary hypercarbia) [19]. Considering this information, it is essential to routinely screen patients at risk of developing acute coronary syndrome or myocardial oxygen supply–demand imbalance and future research should focus on the actual utility of critical care and the potential benefits that arise from continuous heart monitoring that may help reveal the developing pathology in its early stages.

Leaving aside the actual impact of surgery on cardiac pathology, major abdominal operations are frequently associated with fluid overload and problematic postoperative fluid management. Preventive strategies to help reduce the incidence of pulmonary edemas have been proposed and according to the latest evidence in maximal effort cytoreductive surgery it seems that patients with a fluid balance that exceeds 3000 mL have increased odds of developing major complications, whereas patients with a negative fluid balance do not seem to have increased rates of major morbidity [22]. Major factors that seem to influence the possibility of developing severe fluid overload and pulmonary edema include pre-existing heart and lung pathology, increased intraoperative blood loss, rigorous fluid administration and low albumin levels [23]. Goal-directed fluid management significantly limits the possibility of severe cardiopulmonary complications [23] and its implementation is easier in an intensive care establishment using invasive monitoring.

Experience in postoperative care of these patients seems to be very important. Studies have shown that the impact of the volume of operations on the actual outcome of patients that experience cardiovascular complications following major abdominal surgery seems to be extremely significant according to a large retrospective study from the U.S. [24]. Specifically, the results of this large cohort that included almost one million patients from the Nationwide Inpatient Sample indicated that those treated in high volume centers were less likely to develop severe myocardial infraction and had significantly higher odds of surviving from a major cardiovascular incident compared to those treated in low volume centers. It should also be mentioned that certain procedures that seem to be rather common in advanced ovarian cancer, including splenectomy and colectomy, significantly increased the incidence of myocardial infraction (by approximately 50%) and cardiac arrest (2 to 3 times compared to control patients), indicating the need for an intensive postoperative follow-up of these patients.

Deep vein thrombosis (DVT) is a frequent complication that is encountered primarily in ovarian cancer patients with an incidence that may reach approximately 25% of cases [25,26]. The reported incidence of DVT in other gynecological cancers varies with studies focusing on endometrial cancer patients mentioning rates that range from 0.8 up to 8.1% of patients [27], while those that focus on cervical cancer seem to report a frequency between 0.7 and 12.3% [28]. Most common underlying causes of DVT include increasing age, reduced performance status, advanced stage disease as well as medical comorbidities including obesity, cardiopulmonary disease and development of surgical complications [29]. Despite the use of thromboprophylaxis, a significant proportion of them will already present or eventually develop pulmonary embolism (PE) [30]. Considering that half of those events will occur during neoadjuvant chemotherapy [31], and the fact that current guidelines suggest that patients with PE should ideally receive elective surgery 3 months following the thromboembolic event, it becomes reasonable to assume that postoperative intensive care hospitalization is a pre-requisite to surgery as women with gynecological cancer do not have the privilege of long waiting intervals [32]. Therefore, the ability to manage these patients in the critical care setting can help avoid postoperative major complications including cardiovascular incidents as well as bleeding events from anticoagulant therapy, together minimizing the waiting time until surgery. Considering that patients with DVT have an already increased risk of recurrence of the thrombotic events, scoring systems for cancer patients that are hospitalized in critical care facilities have been developed to help predict those at risk, thus, helping the proactive management of this population [33].

### 3.2. Renal Function

Acute kidney injury is commonly encountered in patients undergoing major abdominal surgery. It is directly associated with prolonged hospitalization, and a significant deterioration of the patient’s physiology, which results in increased morbidity and mortality rates [34,35,36,37]. Specific criteria for the definition of acute kidney injury have been established by the Kidney Disease Improving Global Outcomes (KDIGO) that classify it in three stages [38]. The majority of cases with postoperative acute kidney injury is mild/moderate and is manifested with modest increases in serum creatinine [39]. Subtle changes in this latter index should not be misinterpreted; however, previous research suggests that its production may be decreased as a result of reduced muscle creatine content and metabolism, thus, masking the severity of this complication [40]. This can significantly affect the postoperative period as it may result in a need of dialysis, increased risk of infection and even progression to chronic kidney disease [41,42].

Potential predictors of the severity of postoperative renal dysfunction include age, body mass index, the extent of surgical resection, emergent surgical procedures and previous comorbidities, including cardiovascular disease, diabetes mellitus, ascites, mild preoperative renal insufficiency and chronic obstructive pulmonary disease [37,43].

Renal insufficiency is a rather common comorbidity in cancer patients and seems to be multifactorial (Figure 2) [44]. Prerenal causes included dehydration, which may be the result of vomiting, bleeding from the tumor bed or diarrhea, or cardiac dysfunction, which may be owed to hypotension and arrythmia and vascular causes, namely vasculitis. Renal causes may be owed to paraneoplastic features that lead to glomerular dysfunction or from renal metastases that affect the tubular function and the interstitial space. Post-renal causes that may provoke acute renal dysfunction include invasion of the ureters or urinary bladder from the tumor, resulting in obstructive disease and hypercalcemia or hyperphosphatemia that may be the result of chemotherapy and tumor lysis syndrome [45]. It is estimated that approximately one third of cancer patients will develop renal failure [46] and this directly affects their possibility to receive adjuvant treatment, therefore indirectly affecting their survival rates [47].

In gynecological cancer, chemotherapy is considered an indispensable tool for the treatment of advanced stage disease as well as of recurrent tumors. Platinum-based compounds are considered the cornerstone of treatment and are almost entirely eliminated by the urinary tract. Both carboplatin and cisplatin have a molecular mass that exceeds 300 Dalton and may result in significant nephrotoxicity if an appropriate dose reduction is not considered [47]. Specifically, platinum-based compounds seem to impair the function of proximal tubules by promoting DNA damage, apoptosis and inflammation, which results in decreased glomerular filtration [48]. Taxanes, and particularly paclitaxel, which is the predominant drug in this group of patients, rarely disrupt renal function; however, reports of degenerative changes in the renal parenchyma have been suggested even in minimal tolerates doses [49]. Nevertheless, given the rarity of these side effects, the current evidence does not suggest that dose reduction in these regimens is necessary in patients with renal dysfunction [50]. Similarly, other common systemic treatments, including anti-angiogenic factors such as bevacizumab and Poly ADP-Ribose Polymerase inhibitors (PARP) may result in severe renal impairment by reducing the estimated glomerular filtration rate (eGFR) and increasing the rates of proteinuria [51,52]. Little is known about the actual survival rates of gynecological cancer patients that develop renal failure as the available data are extremely limited in the international literature. Cervical cancer patients with obstructive uropathy and hydronephrosis seem to have significantly shorter survival compared to those without renal dysfunction [53]; however, it remains unknown if the placement of a ureteral stent or the use of nephrostomy may benefit their outcomes. In the field of ovarian cancer and endometrial cancer, the paucity of relevant data is observed, indicating the need for future research which may help ameliorate the perioperative outcome of these patients and possibly increase their survival benefit.

### 3.3. Pulmonary Function

Postoperative pulmonary dysfunction is one of the most common issues affecting patients that undergo major abdominal surgery. In a recent study that was conducted by the American College of Surgeons’ National Surgical Quality Improvement Program, researchers observed that approximately 5.8% of the 165,196 patients that were involved suffered from postoperative pulmonary complications, including pneumonia, prolonged intubation support and unplanned intubation [54]. Patients’ performance status, prolonged surgical procedure, ascites, smoking, advanced age (>80 years) and preoperative pulmonary morbidity were significant predictors of major postoperative pulmonary morbidity. Another multicentric study revealed that the same factors were independent predictors of unplanned postoperative intubation [55], indicating the importance of proper preoperative patient counseling and selection for inclusion in an intensive care setting.

The most frequent cause of pulmonary dysfunction in gynecological cancer patients is ascites with or without concurrent pleural effusion. This results in a significant restrictive lung disease which seems to produce functional impairment even in the absence of pre-existing respiratory pathology [56]. Pulmonary metastases are relatively rare in gynecological cancer patients as the majority of cases with advanced stage disease develop lymphatic metastases or metastatic disease through direct spreading to surrounding tissues. Another frequent cause of pulmonary dysfunction is diaphragmatic stripping, which is a procedure that is frequently necessary in advanced ovarian cancer surgery.

It is important to stress out the actual impact of pulmonary dysfunction on cancer survival, as a large registry that was based on 7,846,370 cancer patients indicated that the mortality of those suffering from COPD (chronic obstructive pulmonary disease) reached 261.5/100,000 person-years, which was twice that recorded in the control population [57]. In terms of perioperative survival, one study revealed that patients that experienced pulmonary complications following cytoreductive surgery had an increased risk of 30-day readmission that did not, however, increase the risk of inpatient mortality [58]. It is important to emphasize that, to date, there is no evidence concerning the actual impact of pulmonary dysfunction, either pre-existing or postoperative, on gynecological cancer-related survival. In an article published recently by our institution, we indicated that the deterioration of HCO_3_ in arterial blood gasses retrieved within the first postoperative 48 h, was directly associated with recurrence free survival (*rho* = 0.704, *p* = 0.005) [59]. However, neither the occurrence of postoperative pleural effusion, nor the insertion of a pleural catheter significantly affected the progression free and overall survival of these patients. Nevertheless, it remains important to expand the available knowledge and evaluate if hospitalization in high dependency units actually ameliorates the postoperative outcome of these women.

## 4. Importance and Structure of the Surgical Critical Care Units

With the advancement of surgical skills, the need for comprehensive and individualized patient care is becoming more demanding in our era. The extension of surgical interventions on the tumor load involving the upper abdominal cavity has increased the postoperative complications of patients [60] and observation that seems to be directly related to survival outcomes [61], indicating the need for rigorous perioperative patient management. Moreover, the gain of knowledge in the perioperative management of elderly patients, including octogenarians, also resulted in an increased number of operations during the last 20 years [62,63]. These patients represent a particularly demanding group as physicians have to deal with complex underlying pathology and several cases of malnutrition, which per se predisposes to severe postoperative complications [64,65].

In a previous study based on the National Cancer Database and published in 2019, researchers stated that the surgical complexity score increased in procedures targeting the treatment of ovarian cancer patients, with rates of moderate complexity operations increasing from 28.9% in 2004 to 33.5% in 2015 and for high complexity score procedures from 26.3% to 30% in the same time frame [66]. It is noteworthy to note that in the same study, both the postoperative 30-day mortality as well as the 90-day mortality rates of these patients considerably declined, with a 10% increase in 5-year survival rates (49% from 39.7%). Although this was not directly attributed to the formation of dedicated surgical ICUs, it is already known that the demand for intensive postoperative management of elective procedures has increased overtime, resulting in a percentage that reaches approximately 35% of all ICU admissions in modernized societies [67]. In a worldwide setting the surgical demand of ICU admission following elective surgery is also high, as approximately 10% of patients are immediately admitted to an ICU department, whereas 16% will ultimately need some form of intensive care management to treat their complications [68].

### 4.1. Organization of Critical Care Units

Surgical critical care management should be structured thoroughly to permit state-of-the-art management during the postoperative period. To date, it remains a matter of debate who should be in charge of the postoperative intensive care of patients [69]. Surgeons are the physicians that have established a more direct relationship with the patients that rarely see anesthesiologists or intensivists prior to their admission. The debate among intensivists and surgeons concerning patient care has been previously described [70]; however, to date, almost all published papers focus on philosophical aspects, whereas comparative research with analytic statistics is missing. Moreover, it should be noted that surgical residents seem to be less comfortable with the surgical critical care, therefore, putting into question their abilities as future specialized doctors for the management of these cases [71]. Until further information becomes available, it seems appropriate to mention that thorough communication of the surgical team with the intensive care management team ensures rigorous patient management that allows comprehensive and early initiation of feeding, antithrombotic prophylaxis and patient return to the surgical unit [72]. Coverage of critical care management by intensivists increases the potential of critical care management as it seems to be associated with significantly decreased reductions, with decreased rates of postoperative morbidity, namely the incidence of acute kidney injury and ventilator support, as well as with an important reduction in 30-day mortality [73,74]. The importance of appropriately trained nursing staff is of paramount importance, as studies have indicated that patient safety culture is directly related to the actual outcomes that are expected from critical care units [75]. Most common problems that are expected to arise are related to the frequency of adverse event reporting, therefore indicating the need for multidisciplinary ward rounds, which seem to increase the collaboration and quality of care [76].

While it is relatively easy to define the need of critical care among patients with specific indications, it remains relatively difficult to define the level of postoperative support that is actually needed, which seems to be dependent on several factors that should be evaluated simultaneously. This requires thorough knowledge of the patients’ background risk, assessment of intraoperative parameters that might influence the risk of cardiopulmonary collapse and is also dependent on the actual support from non-ICU critical care facilities (Figure 2).

### 4.2. Intensive Care Units in Gynecologic Oncology

#### 4.2.1. PACU

The post-anesthesia care unit (PACU) is designed as a step-down alternative to the intensive care unit (ICU) and serves as an intermediate station between the operating theater and the surgical ward. It may be often misinterpreted as the recovery room; however, as an establishment PACU is an entirely independent unit which receives patients that undergo intermediate or high complexity score procedures that require an intensive follow-up. In its ideal formation, PACU involves physicians and nursing staff appropriately trained in advanced life support, namely anesthesiologists, intensivists and intensive care nurses [77]. Ideally, patients should be discharged from the PACU within 24 h and advanced to the general ward or a critical care units using evidence-based criteria, such as the Aldrete scoring system [78]. Several factors seem to affect the length of stay in the PACU, and in a recent article researchers observed that increased age, non-partnered status and an ASA (American Society of Anesthesiologists) score ≥3 were the most important predictive factors [79], although the surgical complexity was not considered among the potential predictive variables. It should be noted that PACU is not designed to administer critical care, but to stabilize patients; therefore, prolonged stay should be avoided in frail/unstable patients as previous research suggests that delayed transfer to the ICU may significantly affect patients’ mortality rates [80]. The introduction of the PACU concept in gynecologic oncology has been studied by other researchers as well who observed that patients with advanced ovarian cancer undergoing complex cytoreductive procedures might be mobilized later, compared to patients that were admitted to a pre-PACU, but, nevertheless, had similar hospitalization duration [81]. Research in the field of gynecologic oncology remains, however, extremely scarce and is urgently needed, as evidence from major non-cardiac operations suggests that the contribution of PACU may be extremely important as it seems that it is directly associated with a reduction in the total postoperative length of stay [82,83].

#### 4.2.2. High Dependency Unit

High dependency units (HDUs) are designed for patients that require intensive observation but are not considered candidates for critical care management. The importance of hospitalization in HDUs among patients that underwent major abdominal surgery has been described 20 years ago, when Jones et al. observed fewer cardiorespiratory complications in a series of 192 patients [84]. It should be noted that not all patients require HDU specific interventions, and according to previous researchers it is relatively hard to predict those that actual need to be hospitalized in surgical HDUs [85]. The lack of specific criteria has led to considerable shortage of HDU beds for surgical patients, together diminishing the beneficial effect of the unit [86]. On the other hand, late identification of patients in need of an HDU may considerably affect their treatment, resulting in increased demands of transfer to intensive care units and increased mortality rates [87]. The actual setup of HDUs remains a matter of debate as in several institutions the HDU is a component of the ICU department, whereas in others it is an entirely separate unit. A large multi-institutional study from the UK indicated that stand-alone HDUs have a comparable effectiveness to that of integrated HDUs [88]. This observation is particularly important as it provides substantial evidence towards the actual formation of HDU-dedicated space, without the need for a reduction in ICU beds in facilities that do not currently support surgical high dependency units. In gynecologic oncology, research concerning the necessity of surgical HDUs is extremely limited. In a previous paper published by our team we observed that age was the only preoperative predictor of prolonged HDU hospitalization, whereas bowel resection and increased blood loss were the most likely operative characteristics that could affect physicians’ decision to increase the duration of HDU stay [89].

#### 4.2.3. Intensive Care Unit

The surgical intensive care unit (ICU) is essential in the perioperative management of patients undergoing major abdominal surgery, as despite the low mortality that is ensured during the last years, the number of high risk patients gradually increases, as a result of the aging population and the comorbidities that arise from the obesity epidemic [90]. An increased demand is observed of the surgical ICU is observed over the years, as a result of erroneous patient selection for admission in the ICU, whereas at the same time a significant proportion of patients remains in the ICU beyond the necessary time period [91]. The importance of intensive care support in gynecologic oncology has been studied, although scarcely, and it seems to significantly influence the course of cancer patients. Factors that seem to determine the decision to admit them include poor performance status, hypoalbuminemia, diagnosis of ovarian cancer and extent of debulking surgery as well as the presence of severe comorbidities [92,93,94]. Patients with severe cardiovascular or respiratory pathology often require longer ICU stay [95]. Available research suggests that prolonged surgical ICU hospitalization is associated with poorer survival rates, albeit it remains unknown if structured prehabilitation programs could ameliorate their postoperative course and have an actual impact on survival outcomes [96,97]. Besides surgically treated patients, it is important to denote the importance of intensive care support in cancer patients undergoing chemotherapy as well as those requiring medical treatment for end-of-life symptoms as a considerable amount requires admission to an ICU [98,99,100]. In this particular group of patients, current research suggests clinical trial participation even if chemotherapy is administered in the terminal stage, as rigorous patient management within a critical care facility may ameliorate symptoms and even prolong survival, therefore denoting the importance of ICU quality of care [101]. It is important to evaluate the level of experience of physicians covering intensive care facilities as it is directly related to patient outcomes [102]. Another important factor is ensuring the presence of a surgical intensivist as specialized physicians significantly improve ICU related mortality [103].

Due to the increasing demand of surgical ICUs, the shortage of ICU beds is observed, which in turn results in delays of surgical operations that may affect the course of cancer patients [104,105,106]. Proper evaluation of patients in need of critical care support is of paramount importance and stratification according to their perioperative risk in order to help reduce the proportion of patients that are subject to considerable surgical intervals.

## 5. Criteria for Admission and Discharge from Critical Care Facilities

Despite the significant research that has been undertaken in the field of major abdominal surgery, guidelines concerning the selection of patients for admission in critical care units are still lacking. Similarly, it remains extremely vague which patients should be discharged. In general, criteria for admitting patients in critical care are based on organ failure or on patients with rapidly deteriorating organ function who need rigorous support. The latest ICU Admission, Discharge, and Triage Guidelines that were published in 2016 indicate that ICUs should be reserved for critically ill patients in need of hourly monitoring with invasive methods [107]. These patients require life support for organ failure and often require invasive ventilation, continuous renal replacement therapies, invasive hemodynamic monitoring to direct aggressive hemodynamic interventions, extracorporeal membrane oxygenation, intra-aortic balloon pumps or other forms of critical care. High dependency units are reserved for patients that are unstable or need nursing interventions or laboratory workup frequently (every 2–4 h). These patients often present organ dysfunction, but do not require rigorous interventions such as those mentioned previously. The actual criteria for patient discharge are vaguer and are based on parameters referring to the criteria of admission in the next lower level of care.

Considering the peculiarities of surgical patients it seems reasonable to specifically evaluate the criteria for admission of those submitted in high complexity procedures in critical care facilities. In a recent article, Zampieri et al. suggested that in general, the decision for the admission of surgical patients in a critical care facility should be based on the interplay of several factors including (1) the complexity score of the procedure, (2) patient comorbidities and current status, (3) the risk of development of acute complications (crash risk) and (4) the existence of appropriate care outside the ICU within the hospital (Figure 3) [108]. Moreover, taking into account the latest Delphi consensus that was published by the Royal College of England, physicians should consider that patients with Rnormal temperature, mean arterial pressure, pH, lactate and gas exchange may be safely transferred to the surgical ward, provided that in three consecutive hourly urine volumes the minimum output was >0.5 mL/kg [109]. The university of Heidelberg published strict criteria for ICU and HDU discharge criteria of surgical patients which are based on respirator, cardiocirculatory, neurological parameters as well as nursing care requirements and other variables [110]. Modified criteria are presented in Table 1 which may well fit gynecologic oncology patients undergoing major abdominal surgery.

## 6. Clinical Scenarios of Intensive Care Support in Gynecologic Malignancies

### 6.1. Ovarian Cancer

Ovarian cancer patients usually present with systemic disease that may be considered for upfront major abdominal debulking procedures or neoadjuvant chemotherapy that is followed by interval cytoreduction. The extent of debulking, along with the potential use of chemotherapeutic regimens, renders these patients fragile as they frequently encounter perioperative complications that include severe hemorrhage, postoperative fluid imbalance, transient renal and cardiopulmonary dysfunction as well as potential thromboembolic events. The inclusion of critical care facilities may enhance the recovery of these patients by instituting rigorous postoperative monitoring during the first hours following surgery and also help ameliorate patient symptoms and ensure optimal results in the event of complications.

### 6.2. Endometrial Cancer

Endometrial cancer patients are frequently obese, diabetic and may have cardiovascular and pulmonary comorbidities that require intensive management during the immediate postoperative period. The introduction of critical care facilities can improve the postoperative care of this group of patients during the first postoperative days, as they often require close monitoring to limit the possibility of developing respiratory compromise, hemodynamic instability, and potential surgical complications arising from the technical complexity of surgery in bariatric populations. For cases that develop a recurrent disease that is considered inoperable, intensive care support may also be of use in case of severe bleeding that results in hemodynamic instability.

### 6.3. Vulvar Cancer and Cervical Cancer

The use of critical care facilities in these types of cancer is mainly limited to the actual performance status of patients as well as occurrence of postoperative complications as patients with vulvar cancer rarely develop life-threatening complications following surgery [111], whereas patients with cervical cancer are usually of younger age and usually have an uneventful postoperative course. It should be noted, however, that bleeding event cases that present with advanced stage disease or disease recurrence are not infrequent and in these clinical setting critical care support may improve patient outcomes by allowing close monitoring to help ensure proper resuscitation.

### 6.4. Cost-Effectiveness of Critical Care Support

The actual cost-effectiveness of critical care support in cancer patients is a complex issue as studies show that despite it can be costly, it proves to be life-saving particularly for critically ill cancer patients [112,113]. The actual economic benefit of surgical intensive care units that focus on cancer patients is limited, but proper triage is of particular importance as improper triage may greatly affect the annual costs of healthcare systems [114]. Considering, however, current knowledge, care expenditures seem to increase as the disease progresses and are mainly limited to the end-of-life support [115]. Nevertheless, it should be noted that studies focusing on the economic burden of cancer patients particularly are extremely limited and research has not focused on the actual cost-effectiveness of instituting specific critical care units for cancer patients, let alone patients with gynecologic malignancies.

## 7. Special Considerations

A matter of specific attention that has been underreported and potentially underestimated in the critical care management of surgical patients is the occurrence of post-intensive care syndrome which comprises several entities, including physical, cognitive and mental impairments that occur during the hospitalization period. The associated symptoms that include fatigue, decreased mobility, sleep disturbances and depressive symptoms may significantly alter the course of surgical patients who should be ideally supported by enhanced recovery protocols to help limit surgical morbidity [116]. In this line, the role of dedicated physiotherapists is of paramount importance as even rigorous physiotherapy seems to be of benefit with minimal safety risks [117], provided that an evidence based approach is maintained based on comprehensive indications and recommendations [118]. Similarly, psychiatric assessment of critical care patients seems to particularly important and considering the additive effect of cancer management as well as the accompanying comorbidities that are often present in gynecologic oncology patients, it becomes evident that psychological assessment is essential to help patients cope or even overcome the post-intensive care unit syndrome [119,120].

Another subject which requires further attention is the introduction of patients admitted to HDUs and potentially ICUs in goal-directed enhanced recovery after surgery programs (ERAS). The importance of ERAS in the perioperative care of patients with gynecological cancer has been mentioned in several research papers [121,122,123]. However, certain aspects of ERAS seem to be more challenging primarily due to age-related restrictions, resource barriers, as well as problems arising from the physiology of cancer which may result in significant reduction in the performance status of patients [124]. Nevertheless, by overcoming certain barriers, it seems feasible to implement ERAS even in the most complex cases that are associated with high surgical complexity score procedures [125,126,127]. As most of these cases will more likely require some form of critical care management, it is advisable that modified ERAS pathways are ensured, and that these patients are enrolled in future trials which will assess the feasibility and restrictions of basic concepts of these protocols including (1) the avoidance or downscaling of opioid analgesia, (2) the introduction of early nutrition, (3) the use of copious amounts of protein to promote functional recovery of patients and (4) the promotion of early mobilization which even in our days is frequently perceived as potentially hazardous [128].

## 8. Conclusions

The use of intensive care management units increases in gynecological cancer patients as a result of more aggressive surgery and introduction of systemic treatment in patients with various comorbidities that were opted-out of chemotherapy protocols until recently. Currently, research concerning the actual usefulness of critical care units in gynecologic oncology remains extremely scarce and as the perioperative management of women with cancer advances it seems imperative to investigate the advantages and disadvantages of these units. Together it is essential to establish guidelines that will provide evidence for patient-centered indications for admission, hospitalization services (including enhanced recovery protocols) as well as criteria for discharge. This will render feasible an appropriate training program that will enhance critical care knowledge among gynecologic oncology fellows, thus, substantially improving future patient care.

## Figures and Tables

**Figure 1 cancers-17-02514-f001:**
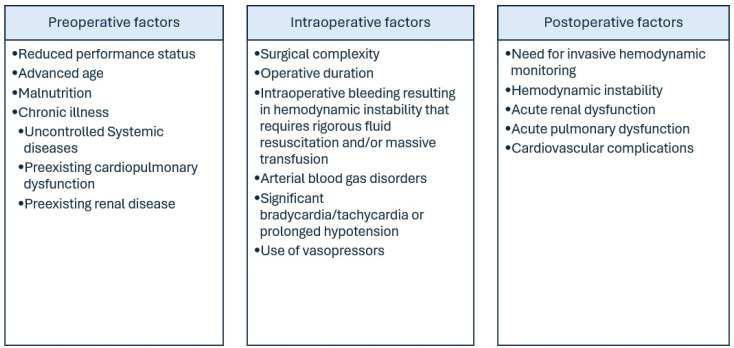
Predictive factors of admission to intensive care facilities.

**Figure 2 cancers-17-02514-f002:**
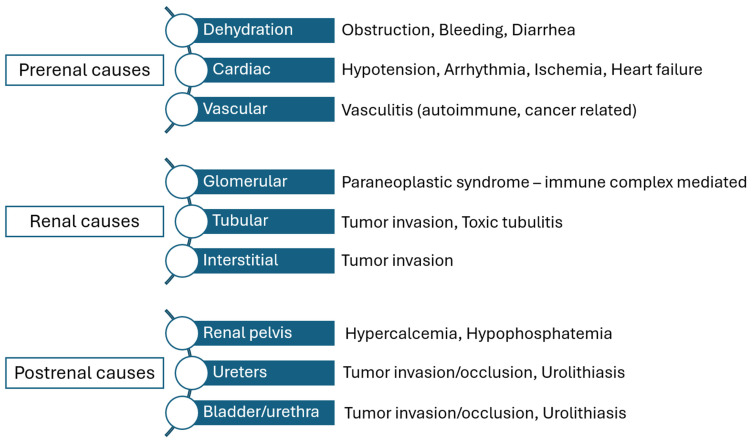
Causes of renal dysfunction in gynecologic oncology patients undergoing surgery and/or systemic therapy.

**Figure 3 cancers-17-02514-f003:**
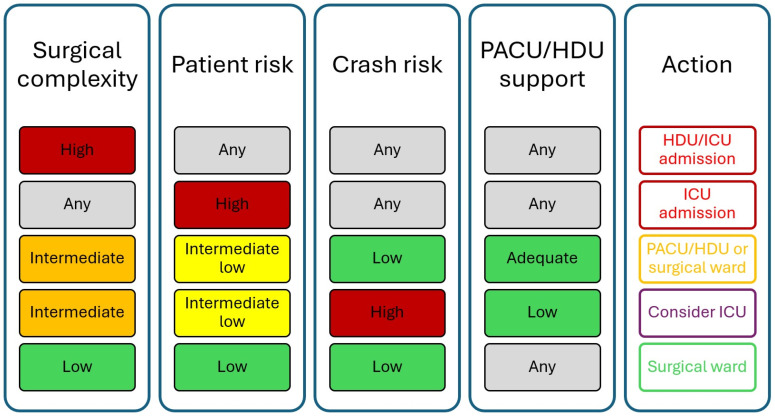
Criteria of admission of surgical patients in intensive care facilities.

**Table 1 cancers-17-02514-t001:** Discharge criteria from intensive care and intermediate care units.

	Intensive Care Unit	High Dependency Unit
Respiratory	Spontaneous respiration	Spontaneous respiration
	<81 O_2_/min	<41 O_2_/min
	CPAP ≤6 h/day	No need for CPAP
Cardiovascular	Noradrenaline infusion ≤0.1 μg/kg/min	No need for vasoactive medication
		No need for invasive monitoring
Urinary system	No need for continuous renal impairment support	Adequate urine output (>0.5 mL/kg/h)
Neurology	No impact on respiratory function	No relevant neurological impairment
Hospital care requirements	No need for nurse–patient care ≤1:2	No need for nurse–patient care ≤1:4
Additional criteria	No lactic acidosis	Serum lactate at normal levels
	No acute bleeding	Confined blood loss (<2 mg/dL/24 h)
		Adequate patient mobilization
		Adequate pain management

## Data Availability

No data available this is a review study with no additional information other than that presented.

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
