# Peer review of "Critical Care Management of Surgically Treated Gynecological Cancer Patients: Current Concepts and Future Directions"

_cancers, 2025, doi:10.3390/cancers17152514_

Round 1
Reviewer 1 Report
Comments and Suggestions for Authors
Pergialiotis et al presented a review on Critical Care Management in Gynecologic Oncology: Current Concepts and Future Directions
The review is well written and the presentation of various preop, intraop and postop factors, discharge criteria from ICU and HDU are good.
Few suggestions are
In figure 1:
In the preoperative period, rather than just systemic disease, uncontrolled systemic diseases would be better predictor of ICU admission.
In the intraoperative factor should include hemodynamic instability, significant intraoperative bradycardiac, tachycardia or prolonged hupotension, and use of vasopressors.
Author Response
We thank the reviewer for the comment.
Figure 1 has been revised in the present draft according to his/her notes.
Sincerely
Reviewer 2 Report
Comments and Suggestions for Authors
Dear Editor,
I carefully read and evaluated the paper “Critical Care Management in Gynecologic Oncology: Current Concepts and Future Directions”
This manuscript addresses a clinically relevant area: the role of critical care management in gynecologic oncology. The authors provide a comprehensive review of perioperative and systemic treatment-related physiological alterations and discuss the role of various levels of critical care facilities, including the PACU, HDU, and ICU. The review is timely and well-organized, offering both theoretical insights and practical considerations.
Point of strengths:
- The manuscript covers an area that is inadequately addressed in the current literature despite its growing clinical importance, given the increasing complexity of gynecologic cancer treatments.
- The discussion is systematically organized by organ system and critical care unit type, facilitating readability and practical application.
- The review is thoroughly referenced, drawing on a wide range of recent and high-quality studies to support its claims.
- The inclusion of evidence-based admission and discharge criteria for critical care units enhances the manuscript’s clinical relevance.
Limitations and Areas for Improvement
- There is insufficient differentiation between types of gynecologic malignancies (e.g., ovarian vs. endometrial vs. cervical cancer), which could add nuance to the discussion.
- Although the abstract briefly mentions hospitalization costs, the manuscript does not explore the economic implications or cost-effectiveness of critical care interventions.
- The review does not propose predictive models or clinical decision-making algorithms, which would enhance its applicability in daily practice
The manuscript is well written, clinically relevant and provides valuable insights into the management of complex gynecologic oncology patients. To increase its applicability, some small changes could be:
- Introduce a clearer stratification of recommendations by tumor type.
- Consider including a schematic flowchart or clinical algorithm for triage and level of care selection.
Author Response
Limitations and Areas for Improvement
- There is insufficient differentiation between types of gynecologic malignancies (e.g., ovarian vs. endometrial vs. cervical cancer), which could add nuance to the discussion.
Authors reply: clinical scenarios of the actual need of critical care in patients with different gynecologic malignancies were included in the present revision
- Although the abstract briefly mentions hospitalization costs, the manuscript does not explore the economic implications or cost-effectiveness of critical care interventions.
Authors reply: there is a large gap of knowledge concerning the cost-effectiveness of critical care interventions in oncology, let alone gynecologic oncology. A relevant section that provides some insight but also mentions this lack of knowledge is presented in the present revision.
- The review does not propose predictive models or clinical decision-making algorithms, which would enhance its applicability in daily practice
Authors reply: we thank the reviewer for this comment. Currently there are no predictive models to support clinical decision making in cancer patients requiring critical care management. We hope that our review will serve as a precursor to future research.
Reviewer 3 Report
Comments and Suggestions for Authors
Dear Authors,
The title of your manuscript is "Critical Care Management in Gynecologic Oncology: Current Concepts and Future Directions." Upon reviewing the entire manuscript, I noticed a discrepancy between the title and the content. The focus of the manuscript is primarily on perioperative critical care rather than the broader scope of gynecologic cancer. Although you briefly mention systemic therapy in a couple of places and refer to critical care for terminally ill patients once, these points do not sufficiently justify the broader title. My suggestion is to keep Critical Care for the Perioperative period to enhance the structure of the manuscript.
Line 31-36: First paragraph on advancing age made me look back on the title of the manuscipt. Please rewrite the first two paragraphs in introduction to bring the broader context of your manuscipt
Figure 1: This is relevant to perioperative Critical Care Only as I have fore mentioned.
2.1: Cardiovascular System: Content needs to be crisp with no repeatitions of thoughts. Line80:I would request the authors to innumerate the impact of abdominal surgery on CVS. And discuss etiology of 1/4 patients dying in postop period from cardiovascular complications. Please emphasize on pre op risk assessment and management once diagnosed. Line 118-123: What is the incidence of DVT and What is the incidence of PE in Gynecologic Cancer and what are the factors associated with higher risk. What is the role of critical care unit in its management.
2.2: Renal Function: Line 126-130: ear authors the first paragraph of Renal Function unit is not crisp and lacks clarity. I appreciate that the etiology of Renal Dysfunction is structured and clear. This Should be done in the Cardiovascular dysfunction also. Figure 2: It Would be interesting to know , Where do you categorize the common chemotherapy: Glomerular/ Tubular Or Interstitial
2.3 Importance and Structure of Surgical Care Units: This unit also is not crisp. Line 246-249: Readers would like to have the most discussion on this point. . I think the Structure of Surgical Care units is PACU, HDO, and ICU which is mentioned in the unit 4.
Unit 3 and 4 would be best merged together . Please cater to the question -How to set up ideal structure in a critical care unit for Gynecological Cancer?
Unit 5 is the most precious. I would suggest to add an author from a intensive care team to impart deeper insights on the ICU management and its role. Otherwise I appreciate Figure 3.
Comments on the Quality of English LanguageDear Authors,
The title of your manuscript is "Critical Care Management in Gynecologic Oncology: Current Concepts and Future Directions." Upon reviewing the entire manuscript, I noticed a discrepancy between the title and the content. The focus of the manuscript is primarily on perioperative critical care rather than the broader scope of gynecologic cancer. Although you briefly mention systemic therapy in a couple of places and refer to critical care for terminally ill patients once, these points do not sufficiently justify the broader title. My suggestion is to keep Critical Care for the Perioperative period to enhance the structure of the manuscript.
Line 31-36: The first paragraph discussing advancing age prompted me to reflect on the title of the manuscript. Please rewrite the first two paragraphs of the introduction to provide a broader context for your manuscript. Figure 1: This figure is relevant only to perioperative critical care, as I have previously mentioned.
**2.1: Cardiovascular System:** The content should be concise, avoiding any repetition of ideas. Line 80: I request that the authors enumerate the impact of abdominal surgery on the cardiovascular system. Additionally, please discuss the etiology of 1/4 of patients who die in the postoperative period due to cardiovascular complications. It is important to discuss preoperative risk assessment and management once these complications are diagnosed.
What is the incidence of DVT and PE in gynecologic cancer, and what factors are associated with a higher risk? What role does the critical care unit play in their management?
2.2: Renal Function: Lines 126-130: The first paragraph of the Renal Function unit is not concise and lacks clarity. I appreciate that the etiology of renal dysfunction is well-structured and clear. A similar approach should be applied to the section on cardiovascular dysfunction. Additionally, in Figure 2, it would be interesting to know how you categorize common chemotherapies: do they fall under Glomerular, Tubular, or Interstitial categories?
**2.3 Importance and Structure of Surgical Care Units:** This section lacks clarity. Lines 246-249: Readers would appreciate more discussion on this point. I believe the structure of surgical care units includes the Post-Anesthesia Care Unit (PACU), the High Dependency Unit (HDU), and the Intensive Care Unit (ICU), which are mentioned in Unit 4.
Units 3 and 4 should be merged to create a cohesive structure. In addition, the question "How can one set up an ideal structure in a critical care unit for gynaecological cancer?" can be addressed. Unit 5 is particularly valuable, and I recommend including an author from the intensive care team to provide deeper insights on ICU management and its role. I have to say, Figure 3 caught my attention! I truly appreciate the insights it brings to the table.
My suggestion is to establish clear aims and objectives, then rewrite the manuscript.
Best Wishes
Author Response
Dear Authors,
The title of your manuscript is "Critical Care Management in Gynecologic Oncology: Current Concepts and Future Directions." Upon reviewing the entire manuscript, I noticed a discrepancy between the title and the content. The focus of the manuscript is primarily on perioperative critical care rather than the broader scope of gynecologic cancer. Although you briefly mention systemic therapy in a couple of places and refer to critical care for terminally ill patients once, these points do not sufficiently justify the broader title. My suggestion is to keep Critical Care for the Perioperative period to enhance the structure of the manuscript.
We thank the reviewer for this remark. In the present revision the manuscript focuses entirely on surgical intensive care management
Line 31-36: First paragraph on advancing age made me look back on the title of the manuscipt. Please rewrite the first two paragraphs in introduction to bring the broader context of your manuscript
Authors reply: according to your first comment the review title and content of this review focuses on the perioperative period in the present revision.
Figure 1: This is relevant to perioperative Critical Care Only as I have fore mentioned.
Authors reply: in the present revision we focus on surgical critical care only.
2.1: Cardiovascular System: Content needs to be crisp with no repeatitions of thoughts. Line80:I would request the authors to innumerate the impact of abdominal surgery on CVS. And discuss etiology of 1/4 patients dying in postop period from cardiovascular complications. Please emphasize on pre op risk assessment and management once diagnosed.
Authors reply: additional information are included in the present revision.
Line 118-123: What is the incidence of DVT and What is the incidence of PE in Gynecologic Cancer and what are the factors associated with higher risk. What is the role of critical care unit in its management.
Authors reply: the incidence of DVT among ovarian cancer patients was mentioned in the previous draft of our review. In the present revision we included data for other gynecological cancers as well. We used the ESMO clinical practice guidelines to mention the potential risk factors that seem to increase the risk for DVT in cancer patients. In our previous draft we reported that “Considering that half of those events will occur during neoadjuvant chemotherapy, and the fact that current guidelines suggest that patients with PE should ideally receive elective surgery 3 months following the thromboembolic event it becomes reasonable to assume that postoperative intensive care hospitalization is a pre-requisite to surgery as women with gynecological cancer do not have the privilege of long waiting intervals.” to denote the importance of critical care units. In the present revision we expanded this section.
2.2: Renal Function: Line 126-130: ear authors the first paragraph of Renal Function unit is not crisp and lacks clarity. I appreciate that the etiology of Renal Dysfunction is structured and clear. This Should be done in the Cardiovascular dysfunction also.
Authors reply: the first paragraph of the renal function section has added information in the present revision.
Figure 2: It Would be interesting to know , Where do you categorize the common chemotherapy: Glomerular/ Tubular Or Interstitial
Authors reply: further data have been introduced about the side effects of common chemotherapy (namely taxane and platinum based compounds)
2.3 Importance and Structure of Surgical Care Units: This unit also is not crisp. Line 246-249: Readers would like to have the most discussion on this point. . I think the Structure of Surgical Care units is PACU, HDO, and ICU which is mentioned in the unit 4. Unit 3 and 4 would be best merged together . Please cater to the question -How to set up ideal structure in a critical care unit for Gynecological Cancer?
Authors reply: units 3 and 4 are merged in the present revision. The question how to structure critical care units was revised to increase the evidence. We would like to note at this point that the evidence in this field is extremely scarce and most articles focus on the philosophical part of who seems to be more “appropriate” for managing sICUs. Comparative studies are missing.
Unit 5 is the most precious. I would suggest to add an author from a intensive care team to impart deeper insights on the ICU management and its role. Otherwise I appreciate Figure 3.
Authors reply: we have already contacted our intensivists to elaborate on Table 1 and Figure 3; however, they did not provide further details as they found those sufficient.
Round 2
Reviewer 3 Report
Comments and Suggestions for Authors
I accept the Manuscipt as it is